# EndoStreamDepth: Temporally Consistent Monocular Depth Estimation for Endoscopic Video Streams

**Hao Li**\*                                              HAO.LI.1@VANDERBILT.EDU
**Daiwei Lu**                                             DAIWEI.LU@VANDERBILT.EDU
**Jiacheng Wang**                                  JIACHENG.WANG.1@VANDERBILT.EDU
**Robert J. Webster III**                         ROBERT.WEBSTER@VANDERBILT.EDU
**Ipek Oguz**                                         IPEK.OGUZ@VANDERBILT.EDU
*Vanderbilt University*

**Editors:** Accepted for publication at MIDL 2026

## Abstract

This work presents **EndoStreamDepth**, a monocular depth estimation framework for endoscopic video streams. It provides accurate depth maps with sharp anatomical boundaries for each frame, temporally consistent predictions across frames, and real-time throughput. Unlike prior work that uses batched inputs, EndoStreamDepth processes individual frames with a temporal module to propagate inter-frame information. The framework contains three main components: (1) a single-frame depth network with endoscopy-specific transformation to produce accurate depth maps, (2) multi-level Mamba temporal modules that leverage inter-frame information to improve accuracy and stabilize predictions, and (3) a hierarchical design with comprehensive multi-scale supervision, where complementary loss terms jointly improve local boundary sharpness and global geometric consistency. We conduct comprehensive evaluations on two publicly available colonoscopy depth estimation datasets, with quantitative results reported on phantom and simulated data that provide ground truth depth. Compared to state-of-the-art monocular depth estimation methods, EndoStreamDepth substantially improves performance, and it produces depth maps with sharp, anatomically aligned boundaries, which are essential to support downstream tasks such as automation for robotic surgery. The code is publicly available at https://github.com/MedICL-VU/EndoStreamDepth.

**Keywords:** Video depth estimation, temporal modeling, state-space models, self-supervised regularization, multi-level supervision

## 1. Introduction

Video depth estimation in monocular endoscopy provides geometric information for downstream tasks, such as 3D reconstruction (Recasens et al., 2021), and supports automation in image–guided and robot–assisted interventions. These applications require depth maps that are accurate, temporally consistent across frames, and available in real time (20 FPS as suggested in (Kasimieh et al., 2025)) so that they can be used in the control loop of surgical robots and support reliable automation.

Existing vision foundation models for monocular depth estimation have achieved state-of-the-art performance (Yang et al., 2024b; Bochkovskii et al., 2025; Chen et al., 2025; Hu et al., 2025; Shao et al., 2025). However, single-frame models may produce flickering and

---

\* Corresponding author

inconsistent depths when they are applied to a video sequence due to the lack of temporal information (Yang et al., 2024b; Bochkovskii et al., 2025). Video-based models (Chen et al., 2025) require processing batches of frames at once for better temporal consistency, which increases latency and makes them less suitable for real-time applications. Diffusion-based depth estimation methods can provide high-quality depth predictions (Hu et al., 2025; Shao et al., 2025), but they also take batches of frames as input and are slow at inference time due to the heavy computational load of diffusion models from iterative denoising steps. Although adapting these foundation models for endoscopic applications could achieve decent performance (Tian et al., 2024; Paruchuri et al., 2024; Cui et al., 2024; Zhou et al., 2025; Hardy et al., 2025), they still suffer from fundamental real-time limitations for video depth estimation, because this setting requires low latency as well as causal, frame-by-frame streaming without waiting for future frames .

A very recent work, FlashDepth (Chou et al., 2025), adapts a large depth foundation model with a temporal Mamba (Dao and Gu, 2024) layer to enable real–time streaming depth estimation with superior performance. However, it is designed for natural indoor and outdoor scenes and does not match the characteristics of endoscopic videos, where near–field lighting, specular reflections, sudden camera movement, rapid rotations, and motion blur cause large appearance changes. These factors limit its performance even with fine–tuning and produce inaccurate depth predictions, particularly with large near-range errors.

In addition, FlashDepth uses only a single $\ell_1$ loss on metric depth for supervision, so far-range errors are penalized more strongly than near-range errors. Due to limited temporal modeling capacity, caused by the lack of temporal consistency regularization and reliance on only a single temporal Mamba module, the predicted depth maps are not stable across frames and exhibit noticeable far-range flickering. Moreover, without an edge-aware supervision term, it fails to produce sharp depth maps in low-texture endoscopic frames.

To overcome these limitations, we propose **EndoStreamDepth**, a streaming monocular depth estimation framework for endoscopic video that produces accurate, temporally consistent, and sharp depth maps. It explicitly models endoscopy-specific geometric and photometric variations and uses comprehensive supervision to achieve robust performance. A hierarchical architecture with multi-level temporal modules that aggregate local and global information, together with a self-supervised temporal regularization term, leverages cross-frame information to maintain consistency throughout the video stream.

For real-world deployment, EndoStreamDepth processes individual frames sequentially, while the multi-level temporal modules maintain the temporal information across arbitrarily long sequences. Our work addresses a key limitation of existing endoscopic video depth methods: they either rely on multi-frame batched input or computationally intensive diffusion processing, both of which introduce significant latency. Our main contributions are:

- We introduce an Endoscopy-Specific Transformation (EST) that models typical geometric and photometric variations in endoscopic video and can be integrated into existing image- and video-based depth foundation models to improve robustness.

- We design a single-frame depth foundation model with a temporal module that leverages inter-frame information for real-time streaming video depth estimation, and we train it with comprehensive supervision to produce accurate, sharp depth maps.

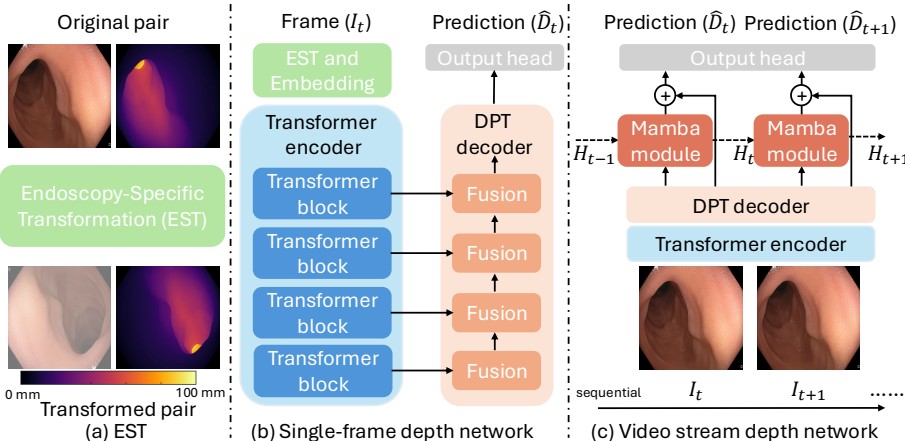

Figure 1: Overview of the **EndoStreamDepth** framework. (a) Endoscopy-specific Transformation (EST) is applied to model typical variations in endoscopy. (b) The single-frame depth network predicts depth map $\hat{D}_t$ from frame $I_t$. (c) The video stream depth network further incorporates Mamba modules that receive hidden states $H_{t-1}, H_t$ to propagate information across frames, improving depth predictions. Frames are processed sequentially (streaming), not simultaneously.

- We propose a hierarchical architecture with multi-level temporal modules, optimized with multi-scale supervision and self-supervised temporal consistency regularization, to produce accurate and temporally consistent depth maps for video streams.

The comprehensive experiments were conducted on publicly available endoscopy metric depth benchmarks, where our quantitative evaluation is conducted on phantom and simulated data with ground truth depth, and the proposed EndoStreamDepth shows superior performance to state-of-the-art depth estimation methods. To the best of our knowledge, this is the first work to stream metric depth estimation for endoscopy with hierarchical temporal modules for arbitrarily long sequences. This work provides a robust, reproducible approach for streaming depth estimation for endoscopic videos.

## 2. Methods

**EndoStreamDepth** (Fig. 1) targets monocular depth estimation from endoscopic video streams. Given an endoscopic video $\{I_t\}_{t=1}^T$, where $I_t \in \mathbb{R}^{H \times W \times 3}$ is the $t^{th}$ RGB frame, it predicts a sequence of depth maps $\{\hat{D}_t\}_{t=1}^T$, with $\hat{D}_t \in \mathbb{R}^{H \times W}$, that are spatially accurate, temporally consistent, with sharp boundaries, and suitable for real-time processing. The framework consists of three components: (1) a single-frame depth backbone with endoscopy-specific transformations to improve robustness, (2) a streaming temporal module based on Mamba that propagates information across frames and is trained with multi-term supervision, and (3) a hierarchical multi-level design that refines depth from local details (edges and fine structures) to global geometry, and enforces a self-supervised temporal consistency loss to stabilize predictions over time.

## 2.1. Single-frame depth network

We first build a single-frame network for accurate depth estimation from endoscopic images. Our single-frame depth network $f_\theta$ (Fig. 1(b)) consists of a Vision Transformer encoder (Oquab et al., 2023) and a DPT decoder (Ranftl et al., 2021). To achieve accurate performance (Li et al., 2025), it adapts pretrained weights from existing large-scale monocular depth foundation models, specifically, DepthAnythingv2 (Yang et al., 2024b). We use ViT-L as the backbone with 24 transformer blocks. The DPT decoder fuses multi-scale features from the encoder using residual blocks (He et al., 2016) to predict the depth map $\hat{D}_t$ from a given frame $I_t$. This single-frame depth network $f_\theta$ serves as the backbone for all subsequent temporal and hierarchical extensions.

Following DepthAnythingv2, we train $f_\theta$ using a scale-invariant logarithmic (SiLog) loss, which is defined as:

$$\mathcal{L}_{\text{si}}(t) = \sqrt{\frac{1}{N} \sum_{i=1}^{N} \left( \log D_t(i) - \log \hat{D}_t(i) \right)^2 - 0.5 \left( \frac{1}{N} \sum_{i=1}^{N} \left( \log D_t(i) - \log \hat{D}_t(i) \right) \right)^2} \tag{1}$$

where $i$ indexes pixels with positive ground truth depth in $D_t$. The loss is scale-invariant and encourages low variance and low bias in the log-depth residuals.

**Endoscopy-Specific Transformation (EST).** Existing depth foundation models (Yang et al., 2024a; Tian et al., 2024) mainly rely on large-scale training data, and do not explicitly consider task-specific variations. For endoscopic depth estimation, task-specific augmentation can be more helpful, but this is rarely discussed in existing work.

To increase the robustness of the depth network, we design a simple yet effective, endoscopy-specific augmentation pipeline that combines geometric transformations (random 90° rotation, horizontal and vertical flips) with photometric perturbations (blur, defocus, brightness/contrast, gamma correction, fog). Examples can be seen in Fig. 5. These transformations are applied to training data before they are fed into the model (Fig. 1(a)), either per frame or per window sequence for frame- and video-based methods.

In clinical procedures, the endoscopic camera is often rotated. Due to the approximately symmetric field of view of the endoscope, the same anatomy may appear in different orientations. The geometric transformations therefore increase the robustness of the network to such viewpoint changes, which are typical in endoscopic applications. The photometric perturbations simulate common endoscopic artifacts, including motion blur, specular reflections from illumination, and smoke and fogging caused by tissue cutting, which frequently occur during surgery. Training the network with these variations improves the robustness of depth estimation from endoscopic images. The details can be found in Appendix A.

## 2.2. Video stream depth network

Building on the baseline $f_\theta$ (Sec. 2.1) and inspired by the recent FlashDepth work (Chou et al., 2025), we extend $f_\theta$ to a video stream depth network for endoscopy (Fig. 1(c)). The network operates in a streaming mode. At time step $t$, the current frame $I_t$ is passed through the $f_\theta$ to obtain a feature map. This feature map, together with the hidden state from the previous step $H_{t-1}$, is then fed into a Mamba module. The Mamba module updates the hidden state and produces a refined feature map for $I_t$, which is decoded by the output head

into the depth map $\hat{D}_t$. The updated hidden state $H_t$ is stored and used at the next time step $t+1$. Over time, this recurrent process propagates temporal information across frames and yields a sequence of depth predictions $\{\hat{D}_t\}$ with improved temporal consistency.

FlashDepth uses only an $\ell_1$ loss on metric depth. This loss penalizes far-range errors more strongly than near-range errors and provides weaker constraints on the global depth structure than the SiLog loss (Eq. 1). In endoscopic video, clinically relevant anatomy often lies at near- and mid-range and requires accurate local geometry and clear anatomical boundaries. We therefore use a SiLog loss together with two auxiliary log-depth losses that emphasize near-range accuracy and boundary sharpness, described below.

**Learning metric depth.** In addition to the temporal module, we supervise $\hat{D}_t$ using a log-domain $\ell_1$ loss defined on the ground truth depth map $D_t$, given by:

$$\mathcal{L}_{\text{metric}}(t) = \frac{1}{N} \sum_{i=1}^{N} \left| \log D_t(i) - \log \hat{D}_t(i) \right| \tag{2}$$

Since $D_t$ is given in physical units (millimeters), minimizing $\mathcal{L}_{\text{metric}}$ encourages both correct metric scale and a coherent global depth shape. Applying a log transform makes the error relative that penalizes $\log(D_t/\hat{D}_t)$ rather than raw differences. This compresses the depth range and reduces the influence of far-range pixels in the supervision, while maintaining accuracy around the near-range region.

**Learning sharp edges.** To sharpen anatomical boundaries, we use an edge-aware loss that emphasizes discrepancies in depth gradients around structure boundaries to preserve thin structures such as lumen borders. Let $G_x(\cdot)$ and $G_y(\cdot)$ denote forward finite differences in the horizontal and vertical directions. The gradient-based edge loss is defined as:

$$\mathcal{L}_{\text{edge}}(t) = \frac{1}{N} \sum_{i=1}^{N} \left( \left| G_x(\log D_t)(i) - G_x(\log \hat{D}_t)(i) \right| + \left| G_y(\log D_t)(i) - G_y(\log \hat{D}_t)(i) \right| \right) \tag{3}$$

This penalizes differences in depth gradients between prediction and ground truth, and thus yields depth maps with sharper, more anatomically aligned boundaries.

### 2.3. Hierarchical multi-level architecture and supervision

The final component of EndoStreamDepth is a hierarchical multi-level design (Fig. 2) that combines temporal Mamba modeling and supervision across scales. Instead of predicting depth at a single resolution, the network produces a pyramid of depth maps and temporal features, which allows it to capture both local fine details and global geometry consistently.

**Multi-level temporal consistency.** We extend the temporal branch (Fig. 1(c)) to four feature levels by applying Mamba modules at multiple spatial resolutions (Fig. 2). Let $l \in \{1, \ldots, 4\}$ index these levels, from finest ($l = 1$) to coarsest ($l = 4$). At each level $l$, the Mamba module contains four Mamba blocks $b \in \{1, \ldots, 4\}$, and each block has a single state-space model (SSM) layer that processes the temporal sequence. This yields, at time $t$, a set of block-wise hidden states $\{h_t^{b,(l)}\}_{b=1}^{4}$. For brevity, we denote the multi-block hidden state by $H_t^{(l)}$ and use $h_t$ to refer to a generic block-wise state when indices are omitted. This

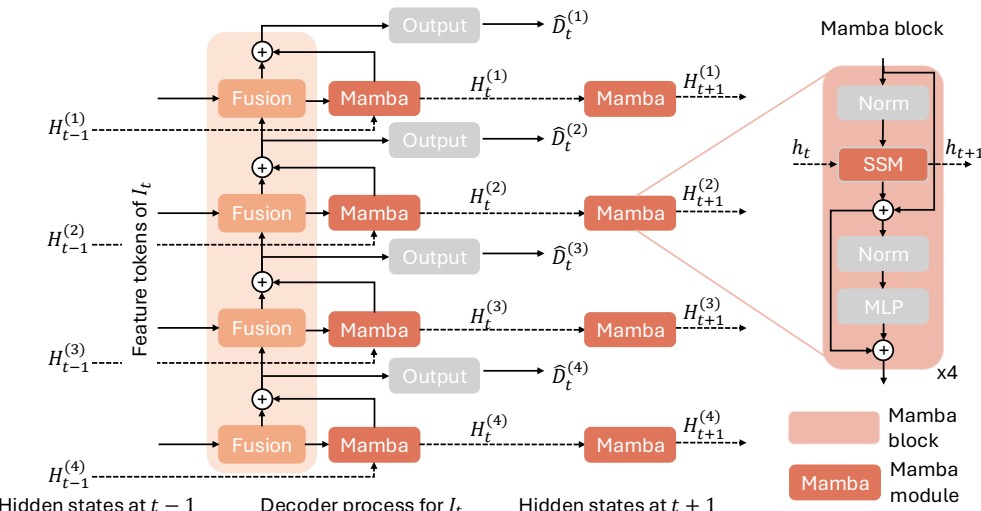

Figure 2: Multi-level temporal Mamba integration within the decoder. For each decoder level $l$, the feature tokens of the current frame $I_t$ are fused with the $l-1$ features and passed through a Mamba module that receives the hidden state $H_{t-1}^{(l)}$ as temporal context. The module outputs an updated hidden state $H_t^{(l)}$, which is propagated to the next frame $I_{t+1}$. The right panel illustrates a single Mamba module implemented as a stack of Mamba blocks with state-space model (SSM) layers, each maintaining a recurrent hidden state $h_t$ that is updated to $h_{t+1}$ at the next time step. For brevity, we denote these internal SSM states by $h_t$ without block indices. They are distinct from the decoder-level states $H_t^{(l)}$, and each SSM layer passes its own hidden state to the corresponding layer at the next time step. Decoder processes for $I_{t-1}$ and $I_{t+1}$ are identical to that for $I_t$ and are omitted.

multi-level temporal modeling allows the network to exploit temporal information at both coarse and fine feature levels, reduces flickering, and provides temporally coherent features.

**Multi-scale deep supervision.** As shown in Fig. 2, the hierarchical decoder produces a pyramid of depth predictions $\{\hat{D}_t^{(l)}\}_{l=1}^4$. The ground-truth depth map $D_t$ is downsampled to these resolutions for supervision, yielding $\{D_t^{(l)}\}_{l=1}^4$. At each level $l$ we apply the same SiLog loss (Eq. 1) to constrain the overall depth shape. Denoting by $\mathcal{L}_{\mathrm{si}}^{(l)}(t)$ the SiLog loss computed between $\hat{D}_t^{(l)}$ and $D_t^{(l)}$, the multi-scale deep supervision is defined as:

$$\mathcal{L}_{\mathrm{ms}}(t) = \sum_{l=1}^4 w_l\, \mathcal{L}_{\mathrm{si}}^{(l)}(t) \tag{4}$$

We set $w_l = 1$, so that all scales contribute equally. Supervising intermediate scales in this way stabilizes training and improves globally consistent depth across the pyramid.

**Self-supervised temporal regularization.** To maintain accuracy and temporal consistency, we introduce an additional temporal regularization loss at the finest scale $\hat{D}_t^{(1)}$. It

penalizes frame-to-frame depth fluctuations with a per-video normalization, so that depth trajectories over time become smoother while spatial details are preserved. For notational simplicity and consistency with Eqs. (1–3), we denote the finest-scale prediction by $\hat{D}_t$.

For a given video with predictions $\{\hat{D}_t\}_{t=1}^{T}$, we first compute a robust per-video normalization. All predicted depths $\hat{D}_t(i)$ across frames and valid pixels are collected to compute a single median $m$ and a mean absolute deviation $a = \frac{1}{TN} \sum_{t=1}^{T} \sum_{i=1}^{N} |\hat{D}_t(i) - m|$. The normalized depth is then given by $\bar{D}_t(i) = (\hat{D}_t(i) - m)/a$, which standardizes the depth scale across videos without altering the spatial structure of the depth maps.

The self-supervised temporal regularization loss is defined over a windowed video as:

$$\mathcal{L}_{\text{temp}} = \frac{1}{(T-1)N} \sum_{t=1}^{T-1} \sum_{i=1}^{N} \left| \bar{D}_{t+1}(i) - \bar{D}_t(i) \right|, \tag{5}$$

which penalizes inconsistent depth changes between frames while remaining invariant to per-video scale and offset, resulting in temporally smoother yet spatially detailed depths.

**Total objective function.** For each training video, EndoStreamDepth is optimized using the four complementary loss terms (Eq. (1–5)). The total training objective is:

$$\mathcal{L}_{\text{total}} = \frac{1}{T} \sum_{t=1}^{T} \left( \mathcal{L}_{\text{ms}}(t) + \mathcal{L}_{\text{metric}}(t) + \mathcal{L}_{\text{edge}}(t) \right) + 0.01\, \mathcal{L}_{\text{temp}} \tag{6}$$

It jointly enforces accurate metric depth, sharp boundaries, multi-scale consistency, and temporal coherence, which are crucial for endoscopic video depth estimation.

## 3. Results

**Phantom colonoscopy depth dataset (C3VD).** This dataset (Bobrow et al., 2023) is a widely used colonoscopy depth benchmark. It contains 22 video sequences that are registered to generate 10,015 frames with paired ground-truth depth maps. The depth values represent distance along $z$-axis of the camera frame and are clamped to the range 0–100 mm. The dataset includes four colon segments: cecum, descending colon, sigmoid colon, and transverse colon. We define two evaluation splits to assess both overall performance and generalization. The first is a domain-shift split, where videos from the transverse colon are held out for testing to evaluate cross-organ generalization. The second is an in-distribution split, where training and test videos are drawn from all four colon segments, following prior work (Paruchuri et al., 2024). Detailed split definitions are provided in Appendix B. This phantom dataset is our primary benchmark and is used for method development.

**Simulated colonoscopy depth dataset (SimCol3D).** It consists of 33 videos with 37,800 frames and paired depths (Rau et al., 2024). This dataset was used in a MICCAI challenge, and we follow the official training and evaluation splits. In our study, SimCol3D is only used for evaluation instead of development.

**Implementation details.** During training, we resize all images to $518 \times 518$ (C3VD) and $476 \times 476$ (SimCol3D), and use the AdamW optimizer with learning rates of $5 \times 10^{-6}$ for encoder and $5 \times 10^{-5}$ for decoder. We train for 15K iterations with a batch size of 4 and a temporal window of 5 frames. All experiments are conducted on an NVIDIA A6000 GPU.

Table 1: Comparison with state-of-the-art on metric depth estimation on the C3VD dataset with the first split (Sec. 3). Distance-based metrics are in mm. The **best** is highlighted for each category. The ablation study is formulated as incremental ablation (each row adds one component), the detailed settings can be viewed in Tab. 10. Gray rows denote the methods for further public benchmarking. The detailed ablation setting can be viewed in Appendix. H. Our model outperforms the compared baselines in every metric.

| Methods | $\delta_1 \uparrow$ | AbsRel$\downarrow$ | SqRel$\downarrow$ | RMSE$\downarrow$ | RMSE log$\downarrow$ | L1$\downarrow$ | F1$\uparrow$ |
|---|---|---|---|---|---|---|---|
| DepthAnything v2 | 0.847 | 0.158 | 0.635 | 3.497 | 0.169 | 2.503 | 0.089 |
| Metric DAv2 * | 0.850 | 0.149 | 0.566 | 3.538 | 0.166 | 2.492 | 0.095 |
| EndoOmni | 0.836 | 0.154 | 0.610 | 3.623 | 0.170 | 2.596 | 0.109 |
| DINOv3 depth | 0.731 | 0.192 | 1.188 | 5.457 | 0.194 | 3.955 | 0.070 |
| FlashDepth ** | 0.730 | 0.188 | 1.046 | 4.989 | 0.190 | 3.780 | 0.116 |
| EndoStreamDepth (Ours) | **0.952** | **0.085** | **0.246** | **2.739** | **0.107** | **1.780** | **0.143** |
| Ablation study (single-frame depth network) | | | | | | | |
| * + EST | 0.948 | 0.109 | 0.402 | 3.081 | 0.122 | 1.928 | 0.114 |
| Ablation study (Video stream depth network) | | | | | | | |
| ** with SiLog | 0.853 | 0.139 | 0.593 | 3.774 | 0.151 | 2.695 | 0.112 |
| + EST | 0.952 | 0.109 | 0.395 | 3.023 | 0.122 | 1.872 | 0.134 |
| + Metric loss | **0.954** | 0.107 | 0.397 | 3.078 | 0.121 | 1.871 | 0.132 |
| + Edge loss | 0.953 | 0.105 | 0.387 | 2.917 | 0.118 | 1.774 | 0.135 |
| + Multi-level temp. | 0.952 | 0.106 | 0.379 | 2.810 | 0.120 | **1.748** | 0.123 |
| + Multi-scale sup. | **0.954** | 0.086 | 0.254 | 2.866 | **0.106** | 1.806 | 0.134 |
| + Temporal reg. (Ours) | 0.952 | **0.085** | **0.246** | **2.739** | 0.107 | 1.780 | **0.143** |

**Compared methods.** We compare to several state-of-the-art depth estimation methods, including a foundation depth model, DepthAnything v2 (Yang et al., 2024b) from natural image domain, EndoOmni (Tian et al., 2024) for medical endoscopic images. In addition, we also compare against the recent video depth estimation method FlashDepth (Chou et al., 2025). Motivated by the strong performance of DINOv3 on monocular depth estimation (Siméoni et al., 2025), we further adapt DINOv3 for medical depth estimation in comparison. All competing methods are implemented using their official code repositories.

For C3VD benchmarking, we compare against PPSNet (Paruchuri et al., 2024), which estimates depth using near-field illumination modeling and a teacher–student framework. In addition, we report the top-performing methods from the official SimCol3D challenge (Rau et al., 2024) on the unseen test sequence (SimCol III) as the comparison.

**Evaluation metrics.** We report standard depth metrics, including absolute relative error (AbsRel), squared relative error (SqRel), root-mean-squared error (RMSE), RMSE in log space (RMSE log), mean absolute error (L1), and the accuracy under threshold ($\delta_1 < 1.25$). To assess spatial detail, we compute the boundary F1 score between predicted and ground-

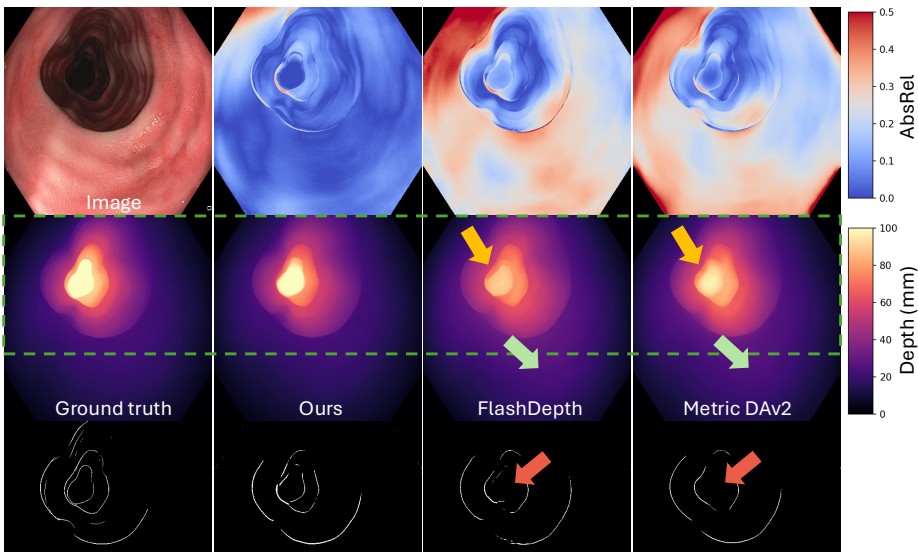

Figure 3: Qualitative results. From top to bottom: AbsRel error maps, predicted depth maps, and edge maps derived from the predicted depth map, cropped to the dashed line for visualization. The yellow and green arrows indicate the far- and near-range errors. Red arrows highlight a defect on the edge maps.

truth depth edges. We additionally report a frame variance metric ($\sigma$) that quantifies the temporal stability of depth predictions across frames (details in Appendix C).

**Overall performance.** The quantitative results of C3VD dataset (split 1) are shown in Tab. 1(top section). The DepthAnything v2 and its metric version (Yang et al., 2024b) show superior performance to other state-of-the-art foundation models for predicting depth maps. These are observed both from relative metrics ($\delta_1$ and AbsRel) and distance metrics (RMSE and L1). However, FlashDepth produces sharper depth maps (F1 score) than the other compared methods. The proposed method, EndoStreamDepth, substantially outperforms these compared methods in global geometry, distances, and anatomical edges. The qualitative results are shown in Fig. 3, where our method exhibits small errors for near- and far-range, and sharp depth, as evidenced by the edge maps (also see Fig. 7).

**Temporal consistency and runtime.** In Fig. 4, we compare the per-video temporal variance $\sigma$ and inference speed in FPS on the C3VD (split 1) for our method and FlashDepth. Across sequences, our method achieves consistently smaller $\sigma$ than FlashDepth on 8/9 videos, indicating that the proposed multi-level temporal modules and temporal regularization effectively suppress frame-to-frame variation. Additional evidence is provided in the ablation study visualization (Fig. 6). As a trade-off, our method yielded lower FPS than FlashDepth (averaged as 24 vs. 36) across all videos. However, it maintains real-time throughput ($> 20$ FPS), which is adequate for medical robot deployment. The details of runtime can be viewed in Appendix. G.

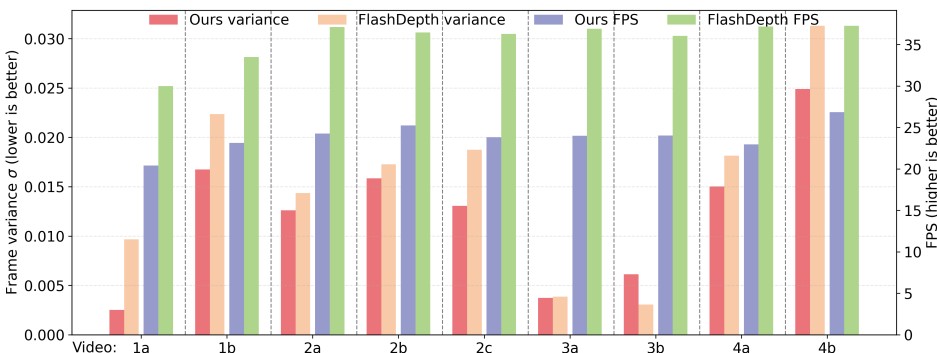

Figure 4: Per-video temporal variance and runtime on the C3VD dataset. For each sequence, bars show the frame-variance score $\sigma$ (left axis) and the inference speed in FPS (right axis) for our method and FlashDepth. Our model has smaller variances than FlashDepth, with the tradeoff of lower FPS.

**Ablation study.** We select both Metric Depth anything v2 and FlashDepth as ablation baseline methods for the single frame-based and video-based depth networks, respectively. The ablation results are displayed in an accumulated way (Tab. 1). We observe that replacing the $\ell_1$ loss with SiLog improves FlashDepth performance. We also found that our proposed simple yet effective Endoscopy-Specific Transformation (EST) dramatically improved these foundation models. Yet, this has not been widely discussed in other endoscopy depth estimation work. The foundation model with EST can serve as a strong baseline.

Additionally, we employed two supervision signals to learn sharp metric depth maps from video streams. The metric loss slightly improved $\delta_1$ with trade-offs on other metrics, while the edge loss further improved all metrics, particularly the distance metrics. With edge loss can introduce flickering in far-range regions because the depth boundaries become too sharp (see Fig. 6). Adding multi-level temporal modules reduces relative-scale (SqRel) and distance errors (RMSE and L1), but F1 decreases slightly. This is because the propagated features are at a coarse scale and lack fine details. Without an explicit constraint to preserve these details, the decoded depth maps become over-smoothed, which reduces F1.

Furthermore, multi-scale supervision substantially improves $\delta_1$, AbsRel, SqRel, and RMSE log, as well as boundary sharpness (F1), with only minor trade-offs in RMSE and L1. Lastly, the proposed self-supervised temporal regularization improves most metrics across relative-scale and distance metrics, with minor decreases in RMSE-log and $\delta_1$. This slight decrease is minor compared to the improvements in relative-scale and distance metrics, which better serve medical robotics applications by ensuring accurate local geometry and globally consistent scale. More importantly, this final version preserves detailed boundary information while maintaining temporal consistency (see Fig. 6).

**EST ablation study.** In Tab. 2, we compare different EST variants built on FlashDepth and report both overall and near-range performance (depth < 3mm). Photometric transformations alone degrade performance, whereas geometric transformations improve results across metrics. Importantly, full EST yields the largest improvements in the near range, no-

Table 2: Ablation study on EST components. Distance-based metrics are in mm. "near" denotes the near-range region < 3mm. The **best** is highlighted for each category. Full EST achieves the best results except for AbsRel.

| Methods | $\delta_1 \uparrow$ | $\delta_{1,\text{near}} \uparrow$ | RMSE↓ | $\text{RMSE}_{\text{near}} \downarrow$ | L1↓ | $\text{L1}_{\text{near}} \downarrow$ | AbsRel↓ |
|---|---|---|---|---|---|---|---|
| FlashDepth with SiLog | 0.853 | 0.831 | 3.774 | 2.789 | 2.695 | 2.197 | 0.139 |
| + photometry only | 0.835 | 0.821 | 4.868 | 3.180 | 3.160 | 2.336 | 0.156 |
| + geometry only | 0.922 | 0.911 | 3.083 | 2.010 | 1.962 | 1.521 | **0.099** |
| + full EST | **0.952** | **0.946** | **3.023** | **1.715** | **1.872** | **1.284** | 0.108 |

Table 3: Ablation study of multi-level temporal Mamba variants. L1 denotes the finest level. "F-Var." represents the frame variance metric. Bold denotes the **best**. The full hierarchy achieves the best AbsRel, RMSE, F1, and temporal stability.

| Methods | L1 | L2 | L3 | L4 | $\delta_1 \uparrow$ | AbsRel↓ | RMSE↓ | L1↓ | F1↑ | F-Var.↓ |
|---|---|---|---|---|---|---|---|---|---|---|
| Mid-level only | | ✓ | ✓ | | 0.952 | 0.086 | 2.996 | 1.869 | 0.117 | 0.0129 |
| Fine only | ✓ | | | | 0.953 | 0.105 | 2.917 | **1.774** | 0.135 | 0.0137 |
| Fine+mid hierarchy | ✓ | ✓ | | | **0.954** | 0.087 | 2.975 | 1.817 | 0.124 | 0.0124 |
| Fine+mid hierarchy | ✓ | ✓ | ✓ | | 0.946 | 0.087 | 2.902 | 1.846 | 0.134 | 0.0124 |
| Full hierarchy | ✓ | ✓ | ✓ | ✓ | 0.952 | **0.085** | **2.739** | 1.780 | **0.143** | **0.0123** |

tably reducing $\text{RMSE}_{\text{near}}$ from 2.789 to 1.715mm. This improvement is especially critical for surgical robotics, where accurate near-range depth supports safe instrument–tissue interaction, indicating that the combined transformations better model near-range perturbations. Since EST is a modality-agnostic stochastic transformation with no learned parameters, it can potentially be directly applied to other endoscopic imaging modalities.

**Temporal analysis.** Tab. 3 shows the placement of temporal Mamba modules across feature pyramid levels (L1=finest, L4=coarsest). Placing Mamba only at mid-levels (L2-L3) produces the worst F1 score due to missing local boundary information. Using only the finest level (L1) preserves detailed local context (L1, F1) but lack of global detail, leading to higher AbsRel and worst temporal stability (F-Var.). Gradually adding coarser level Mamba further improves the AbsRel and gradually improves RMSE and F1. Our full 4-level design captures complementary information across scales, i,e, coarse levels for global temporal context and fine levels for boundary sharpness, achieving the best RMSE, F1 and temporal stability, while maintaining comparable performance for $\delta_1$ and L1. The improvements between last two rows in Tab. 3 indicate the effectiveness of adding coarsest temporal Mamba module.

**Window size analysis.** We analyze the effect of temporal window size on depth estimation quality. As shown in Tab. 4, a shorter window yields higher frame variance, likely due to limited temporal context for enforcing consistency. Increasing the window to 10 reduces

Table 4: Ablation study on temporal window size. "Frame Var" denotes frame variance. Recommended achieves the best results across metrics except for AbsRel.

| Methods | $\delta_1 \uparrow$ | AbsRel$\downarrow$ | RMSE$\downarrow$ | L1$\downarrow$ | F1$\uparrow$ | Frame Var.$\downarrow$ |
|---|---|---|---|---|---|---|
| Window=3 | 0.948 | 0.090 | 2.945 | 1.907 | 0.131 | 0.0142 |
| Window=5 (recommended) | **0.952** | 0.085 | **2.739** | **1.780** | **0.143** | **0.0123** |
| Window=10 | 0.948 | **0.084** | 2.785 | 1.790 | 0.106 | 0.0125 |

Table 5: Benchmarking comparison on C3VD and SimCol. * denotes the methods that solely rely on supervised learning, while ** indicates the model is trained with additional data with self-supervised learning (SSL). 1-4 represents the challenge $1^{st}$ to $4^{th}$ place. The **best** is highlighted. EndoStreamDepth outperforms the compared methods, except for the absolute relative error in C3VD.

| | C3VD split 2 | | | | SimCol III | | |
|---|---|---|---|---|---|---|---|
| Methods | AbsRel$\downarrow$ | SqRel$\downarrow$ | RMSE$\downarrow$ | Methods | L1$\downarrow$ | RMSE$\downarrow$ | AbsRel$\downarrow$ |
| LightDepth | 0.078 | 1.81 | 6.55 | CVML[1] | 0.099 | 0.141 | 0.025 |
| NormDepth+ ** | 0.155 | 1.53 | 7.51 | MIVA[2] | 0.107 | 0.163 | 0.025 |
| PPSNet-Teacher* | 0.053 | 0.15 | 2.15 | EndoAI[3] | 0.111 | 0.168 | 0.028 |
| PPSNet-Student** | **0.049** | 0.14 | 2.06 | IntuitiveIL[4] | 0.167 | 0.233 | 0.047 |
| Ours-frame* | 0.077 | 0.27 | 1.74 | Ours-frame | 0.099 | 0.140 | 0.028 |
| Ours-video* | 0.052 | **0.11** | **1.72** | Ours-video | **0.087** | **0.126** | **0.023** |

frame variance but degrades boundary sharpness (lower F1). A window size of 5 provides the best overall trade-off, achieving the lowest frame variance and the highest F1.

**Benchmarking.** To validate the effectiveness of our model, in Tab. 5 we compare it with two public benchmarks for depth estimation using C3VD and SimCol3D. The results of our single-frame network with the proposed EST are comparable to those of the top-performing comparison methods. The proposed EndoStreamDepth achieved the best results except for the AbsRel in C3VD. This suggests our model shows good generalizability between endoscopic datasets. These benchmarking comparisons also suggest that our methods, either frame- or video-based, can be used as robust methods for endoscopy depth estimation.

## 4. Conclusion

In this paper, we propose EndoStreamDepth, a monocular depth estimation framework for endoscopic video that produces accurate, temporally consistent, and sharp depth maps while maintaining real-time throughput. Our method outperforms state-of-the-art baselines on public depth datasets. More importantly, EndoStreamDepth addresses a key limitation of current methods for real-time applications, which rely on batched multi-frame processing for endoscopic video depth estimation. By processing frames as a stream, EndoStreamDepth reduces latency and satisfies real-time requirements. Future work will extend to other applications of robot-assisted endoscopic intervention and further leverage proprioception (Jordan et al., 2025) to improve performance in areas with relatively larger errors.

## Acknowledgments

Research reported in this publication was supported by the Advanced Research Projects Agency for Health (ARPA-H) under Award Number D24AC00415-00. The ARPA-H award provided 90% of total costs with an award total of up to $11,935,038. The content is solely the responsibility of the authors and does not necessarily represent the official views of ARPA-H. This work was also supported in part by the National Institutes of Health (R21DK133742) and Vanderbilt Institute for Surgery and Engineering (VISE) Seed Grant. Daiwei Lu is supported by NIH F31DK143735-01.

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

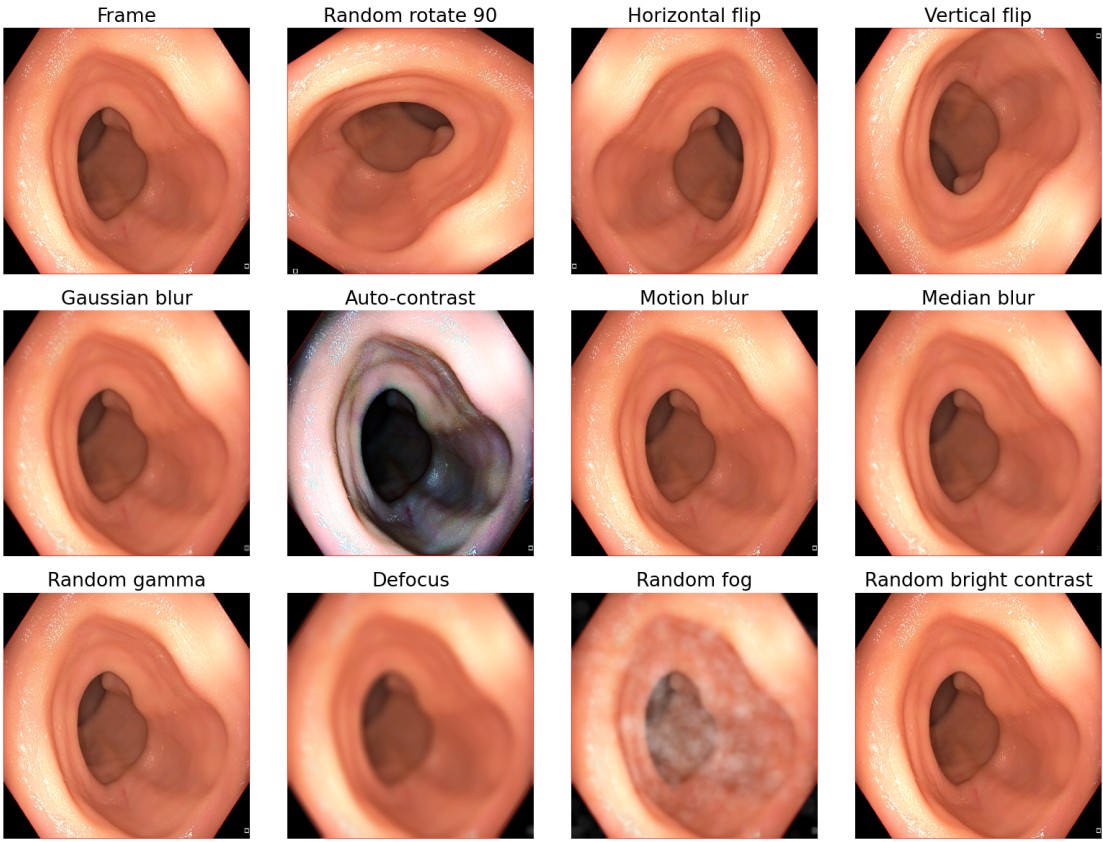

Figure 5: The proposed EST for endoscopic depth estimation.

## Appendix A. EST Illustration

Fig. 5 shows qualitative examples of the individual image transformations used in the proposed EST pipeline. The first panel shows the input frame. The next three panels apply geometric transformations (random 90° rotation, horizontal flip, vertical flip), which mimic camera roll and viewpoint changes that commonly occur during endoscopic procedures. The remaining panels show photometric perturbations, including Gaussian blur, auto-contrast, motion blur, median blur, random gamma, defocus, random fog, and random brightness/contrast. These photometric perturbations approximate typical appearance changes in real procedures, such as variations in exposure, illumination falloff, camera defocus, motion-induced blur, and occlusions from smoke or fog. During training, these transformations are sampled stochastically and applied to the RGB frame (and corresponding depth and masks), so that the depth network is exposed to a wide range of endoscopy-specific appearance variations.

We avoid other geometric warps, such as affine transformations, since the required interpolation on depth maps can introduce artifacts and corrupt the ground truth.

Table 6: C3VD (split 1). See (Bobrow et al., 2023) for dataset details.

| Sequence | Texture | Video | Frames | Set |
|---|---|---|---|---|
| Cecum | 1 | a | 276 | Train |
| Cecum | 1 | b | 765 | Train |
| Cecum | 2 | a | 370 | Train |
| Cecum | 2 | b | 1120 | Train |
| Cecum | 2 | c | 595 | Train |
| Cecum | 3 | a | 730 | Train |
| Cecum | 4 | a | 465 | Train |
| Cecum | 4 | b | 425 | Train |
| Sigmoid | 1 | a | 800 | Train |
| Sigmoid | 2 | a | 513 | Train |
| Sigmoid | 3 | a | 610 | Train |
| Sigmoid | 3 | b | 536 | Train |
| Descending | 4 | a | 148 | Train |
| Transcending | 1 | a | 61 | Test |
| Transcending | 1 | b | 700 | Test |
| Transcending | 2 | b | 102 | Test |
| Transcending | 2 | c | 235 | Test |
| Transcending | 3 | b | 214 | Test |
| Transcending | 4 | b | 595 | Test |
| Transcending | 2 | a | 194 | Test |
| Transcending | 3 | a | 250 | Test |
| Transcending | 4 | a | 384 | Test |

## Appendix B. C3VD Dataset Splits.

The dataset splits used in our C3VD experiments are summarized in Tab. 6 and Tab. 7, which report the train/test configurations for split 1 and split 2, respectively. Split 1 is primarily used for method development and for validating cross-structure performance, whereas split 2 follows the benchmarking of prior work (Paruchuri et al., 2024) and evaluates the generalizability of the methods.

Table 7: C3VD (split 2). P1 and P2 denote the first and second halves of the total frames, respectively. See (Bobrow et al., 2023) for dataset details.

| Sequence | Texture | Video | Frames | Set |
|---|---|---|---|---|
| Cecum | 1 | b | 765 | Train |
| Cecum | 2 | b | 1120 | Train |
| Cecum | 2 | c | 595 | Train |
| Cecum | 4 | a | 465 | Train |
| Cecum | 4 | b | 425 | Train |
| Sigmoid | 1 | a | 800 | Train |
| Sigmoid | 2 | a | 513 | Train |
| Sigmoid | 3 | b | 536 | Train |
| Transcending | 1 | a | 61 | Train |
| Transcending | 1 | b | 700 | Train |
| Transcending | 2 | b | 102 | Train |
| Transcending | 2 | c | 235 | Train |
| Transcending | 3 | b | 214 | Train |
| Transcending | 4 | b | 595 | Train |
| Descending P1 | 4 | a | 74 | Train |
| Cecum | 1 | a | 276 | Test |
| Cecum | 2 | a | 370 | Test |
| Cecum | 3 | a | 730 | Test |
| Sigmoid | 3 | a | 610 | Test |
| Transcending | 2 | a | 194 | Test |
| Transcending | 3 | a | 250 | Test |
| Transcending | 4 | a | 384 | Test |
| Descending P2 | 4 | a | 74 | Test |

## Appendix C. Evaluation Metrics

Let $D \in \mathbb{R}^{H \times W}$ and $\hat{D} \in \mathbb{R}^{H \times W}$ denote the ground truth and predicted depth maps, respectively. Let $\Omega$ be the set of valid pixels and $N = |\Omega|$. For a pixel index $i \in \Omega$, we write $D_i$ and $\hat{D}_i$ for the corresponding depth values.

**Pixelwise depth errors.** We use the following standard depth metrics:

$$\text{AbsRel} = \frac{1}{N} \sum_{i \in \Omega} \frac{|D_i - \hat{D}_i|}{D_i}, \tag{7}$$

$$\text{SqRel} = \frac{1}{N} \sum_{i \in \Omega} \frac{(D_i - \hat{D}_i)^2}{D_i}, \tag{8}$$

$$\text{RMSE} = \sqrt{\frac{1}{N} \sum_{i \in \Omega} (D_i - \hat{D}_i)^2}, \tag{9}$$

$$\text{RMSE}_{\log} = \sqrt{\frac{1}{N} \sum_{i \in \Omega} \left( \log D_i - \log \hat{D}_i \right)^2}, \tag{10}$$

$$\text{L1} = \frac{1}{N} \sum_{i \in \Omega} |D_i - \hat{D}_i|. \tag{11}$$

**Threshold accuracy.** The threshold accuracy $\delta_1$ with factor 1.25 is defined as

$$\delta_1 = \frac{1}{N} \sum_{i \in \Omega} \mathbf{1} \left[ \max\left( \frac{D_i}{\hat{D}_i}, \frac{\hat{D}_i}{D_i} \right) < 1.25 \right], \tag{12}$$

which measures the percentage of pixels whose predicted depth is within a factor of 1.25 of the ground-truth depth.

**Boundary F1 score.** We follow the recent depth estimation work (Bochkovskii et al., 2025) to evaluate boundary sharpness Let $M_i \in \{0, 1\}$ be a validity mask (we set $M_i = 1$ only where $D_i$ is valid and $D_i > 0$). We consider 4-connected neighbor pairs

$$\mathcal{N} = \big\{ (i, j) \mid i \text{ and } j \text{ are horizontal or vertical neighbors}, M_i M_j = 1 \big\}.$$

For a ratio threshold $t > 1$, a neighbor pair $(i, j) \in \mathcal{N}$ is marked as a depth boundary in the ground truth if the two depths differ by more than a factor $t$:

$$B_t(i, j) = \begin{cases} 1, & \text{if } \max\left( \dfrac{D_i}{D_j}, \dfrac{D_j}{D_i} \right) > t, \\ 0, & \text{otherwise}, \end{cases} \tag{13}$$

and we define $\hat{B}_t(i, j)$ analogously using $\hat{D}$.

The precision and recall of predicted boundaries at threshold $t$ are

$$P_t = \frac{\sum_{(i,j) \in \mathcal{N}} B_t(i, j) \, \hat{B}_t(i, j)}{\sum_{(i,j) \in \mathcal{N}} \hat{B}_t(i, j)}, \qquad R_t = \frac{\sum_{(i,j) \in \mathcal{N}} B_t(i, j) \, \hat{B}_t(i, j)}{\sum_{(i,j) \in \mathcal{N}} B_t(i, j)}, \tag{14}$$

and the boundary F1 score is

$$\text{F1}(t) = \frac{2P_t R_t}{P_t + R_t}. \tag{15}$$

We obtain a scale-invariant boundary score by aggregating over multiple thresholds. Let $\{t_k\}_{k=1}^N$ be $N$ thresholds uniformly spaced in $[t_{\min}, t_{\max}]$ and let $w_k$ be weights proportional to the threshold:

$$\text{F1}_{\text{SI}} = \sum_{k=1}^N w_k \, \text{F1}(t_k), \qquad w_k = \frac{t_k}{\sum_{\ell=1}^N t_\ell}. \tag{16}$$

In our experiments we use $t_{\min} = 1.05$, $t_{\max} = 1.15$, and $N = 10$. We're not following (Bochkovskii et al., 2025) to use a maximum range of 1.25 because endoscopic scenes are dominated by smooth, low-texture surfaces where depth discontinuities are often relatively small. Larger ratios, such as 1.25, would not produce enough edge information. Fig. 7 shows edge maps with $t = 1.01$.

**Frame variance.** For each video sequence, we measure temporal variance of the predicted metric scale. For each frame, we compute the optimal global scale factor $s_t \in \mathbb{R}$ that best aligns $\hat{D}_t$ to $D_t$ in the least-squares sense:

$$s_t = \arg\min_s \sum_{i \in \Omega_t} \left(s \, \hat{D}_t(i) - D_t(i)\right)^2 = \frac{\sum_{i \in \Omega_t} \hat{D}_t(i) \, D_t(i)}{\sum_{i \in \Omega_t} \hat{D}_t(i)^2 + \varepsilon}, \tag{17}$$

with a small $\varepsilon$ to avoid division by zero. Given the sequence of scale factors $\{s_t\}_{t=1}^T$, we define the frame consistency score as the standard deviation of these scales:

$$\sigma = \sqrt{\frac{1}{T} \sum_{t=1}^T (s_t - \bar{s})^2}, \qquad \bar{s} = \frac{1}{T} \sum_{t=1}^T s_t. \tag{18}$$

Lower values of $\sigma$ indicate a more temporally stable metric scale (less frame-to-frame flicker).

Table 8: Video-wise statistical evaluation on the C3VD split 1 test set. **Bold** indicates the best mean result. *: $p < 0.05$, ***: $p < 0.001$ (paired t-test vs. both baselines).

| Metrics | Metric DAv2 + EST | FlashDepth + SiLog + EST | EndoStreamDepth |
|---|---|---|---|
| $\delta_1 \uparrow$ | 0.948±0.052 | **0.952**±0.046 | **0.952**±0.047 |
| AbsRel↓ | 0.109±0.025 | 0.109±0.021 | **0.085**±0.021*** |
| SqRel↓ | 0.402±0.135 | 0.395±0.100 | **0.246**±0.109*** |
| RMSE↓ | 3.081±0.528 | 3.023±0.552 | **2.739**±0.442* |
| RMSE log↓ | 0.122±0.025 | 0.122±0.021 | **0.107**±0.023*** |
| L1↓ | 1.928±0.362 | 1.872±0.303 | **1.780**±0.300* |
| F1↑ | 0.114±0.045 | 0.134±0.069 | **0.143**±0.059 |

## Appendix D.  Statistical Significance Analysis

We have updated Tab. 8 to report the video-wise mean mean ± std. across the 9 test sequences, and we additionally conduct paired t-tests for statistical significance. The proposed EndoStreamDepth achieves statistically significant improvements over both baselines on 5 of 7 metrics: AbsRel ($p < 0.001$), SqRel ($p < 0.001$), RMSE ($p < 0.05$), RMSE log ($p < 0.001$), and L1 ($p < 0.05$). In addition, our method shows lower variance on key metrics (AbsRel, RMSE, L1), indicating more robust performance across different test sequences.

## Appendix E.  Qualitative Temporal Stability

Fig. 6 compares two ablations from Tab. 1: "+ Edge loss" (without temporal modules) and "+ Temporal reg." (with the multi-level temporal module and self-supervised temporal regularization). Compared to "+ Edge loss", "+ Temporal reg." produces more temporally consistent depth predictions and noticeably reduces frame-to-frame flickering, especially under rapid camera motion reversals (red arrows).

**Video-level quantitative.**    To complement the qualitative comparison, we report metrics on the same video (2b) shown in Fig. 6 using the evaluation protocol in Tab. 1. Adding temporal regularization improves $\delta_1$ from 0.984 to 0.986 (+0.20%), reduces AbsRel from 0.661 to 0.653 (+1.21%), and reduces RMSE from 2.65mm to 2.59mm (+2.26%). The frame variance also decreases from 0.00209 to 0.00127 (+39.23%), further supporting improved temporal stability.

**Temporal module choice.**    We adopt a Mamba-based streaming temporal module to meet real-time, per-frame inference with a persistent state. This design follows FlashDepth, which reports supplementary comparisons among lightweight temporal alternatives and finds that vanilla Mamba is sufficient in most cases. Building on this established streaming backbone, our contributions focus on endoscopy-specific robustness (EST), multi-level temporal integration, and comprehensive supervision to further improve the performance and reduce flickering.

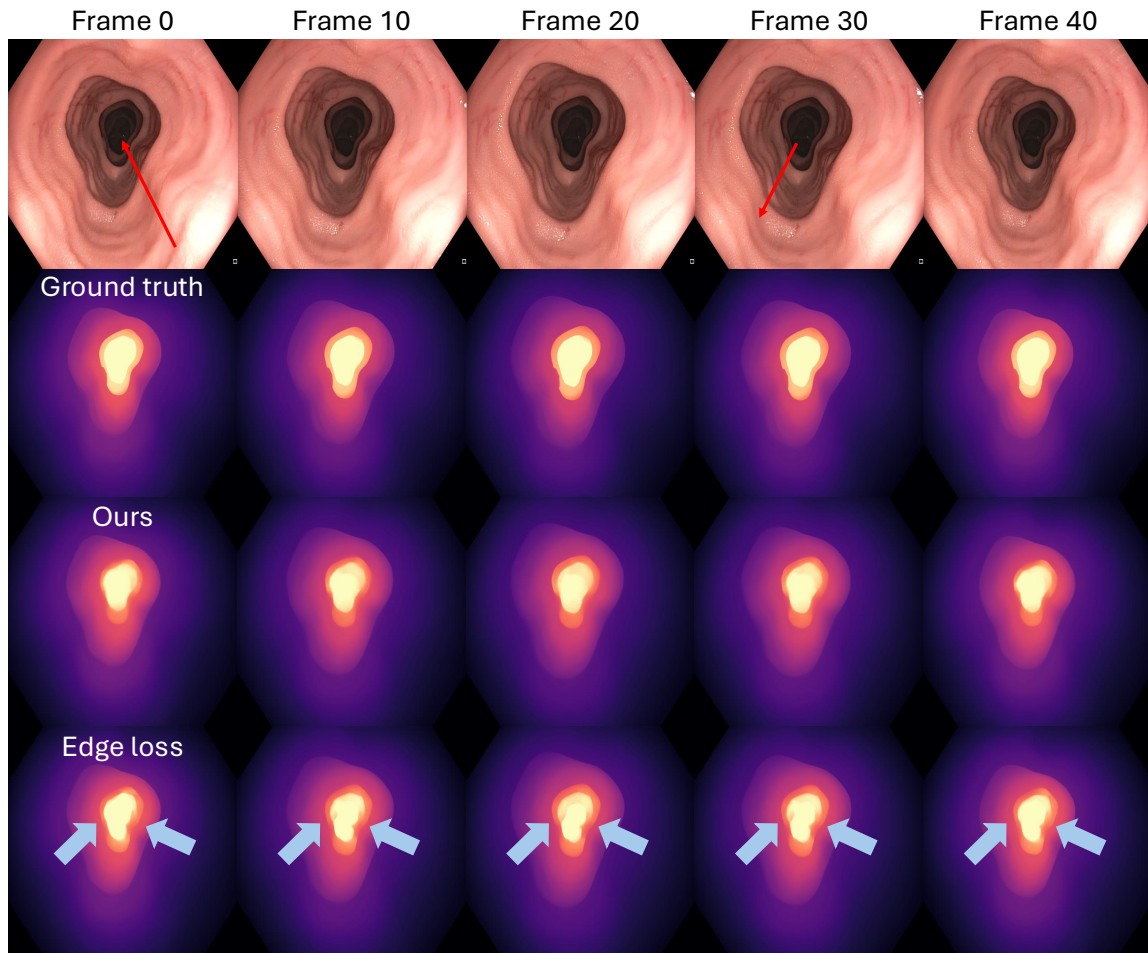

Figure 6: Temporal qualitative comparison between the two ablations in Tab. 1. "+ Edge loss" refers to the "+ Edge loss" row (edge loss included) in Tab. 1, i.e., without the multi-level temporal modules or temporal regularization. Temporally inconsistent predictions are marked by blue arrows. Red arrows indicate the camera motion, moving forward and then backward.

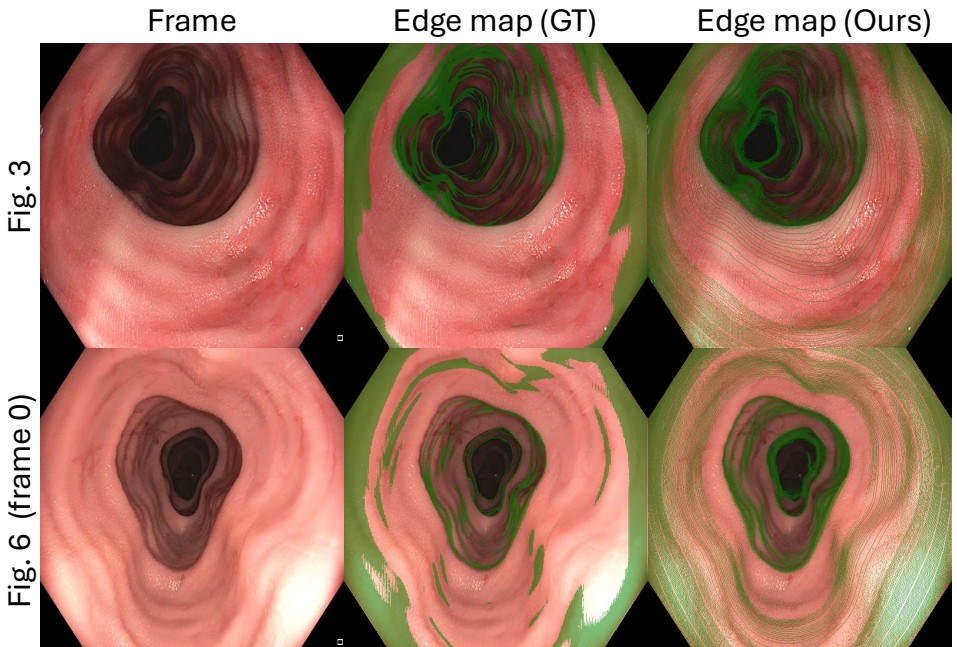

Figure 7: The edge maps (green) overlaid on frames derived from ground truth and our prediction depth maps. The left side bar indicates the frames in Fig. 3 and Fig. 6. The edge maps suggest that our methods can capture even slight changes in low-texture regions of the scene. The edge maps were eroded with a 3-pixel kernel to exclude the field-of-view border. The edge maps are derived with $t = 1.01$. The details of the edge maps are in Appendix. C.

## Appendix F. Sharp Depth Map

Fig. 7 shows the qualitative edge maps derived from ground truth and our predicted depth maps.

Table 9: Runtime and memory usage versus video length. The inference speed remains close with different video lengths

| Video Length (frames) | Per-Frame Latency (ms) | FPS | Peak GPU Memory |
|---|---|---|---|
| 100 | 40.2 | 24.9 | 2.90 GB |
| 500 | 40.0 | 25.0 | 4.50 GB |
| 1000 | 40.3 | 24.8 | 6.50 GB |
| 3000 | 40.7 | 24.6 | 14.50 GB |

## Appendix G. Runtime Analysis

Tab. 9 reports latency and memory benchmarks at $518 \times 518$ resolution on a single NVIDIA RTX A6000 GPU. We use bfloat16 mixed precision and `torch.compile` for efficient PyTorch inference. We skip the first 5 frames for warmup and measure steady-state performance on the remaining frames. We additionally benchmark up to 3000 frames, exceeding the maximum sequence length in our dataset (700 frames, see Tab. 6) to assess scalability.

**Constant inference speed.** Per-frame latency remains $\sim 40$ ms ($\sim 25$ FPS) across video lengths, meeting real-time requirements ($\geq 20$ FPS) and indicating low computational overhead of the Mamba-based streaming temporal module.

**Predictable memory scaling.** Peak GPU memory increases approximately linearly with video length. In our setting, it consists of a fixed base cost (model weights and buffers) plus an incremental cost of $\approx 4$ MB per frame for the temporal state, enabling straightforward memory budgeting for long sequences.

## Appendix H. Ablation Setting

Table 10: Ablation settings (enabled components) for EndoStreamDepth on C3VD (split 1).

| Methods | $L_1$ | $L_{\mathrm{SiLog}}$ | EST | $L_{\mathrm{metric}}$ | $L_{\mathrm{edge}}$ | Multi-level temp. | $L_{\mathrm{ms}}$ | $L_{\mathrm{temp}}$ |
|---|---|---|---|---|---|---|---|---|
| *Single-frame depth network* | | | | | | | | |
| Metric DepthAnything v2 | | ✓ | | | | | | |
| Metric DepthAnything v2 + EST | | ✓ | ✓ | | | | | |
| *Video-stream depth network* | | | | | | | | |
| FlashDepth | ✓ | | | | | | | |
| FlashDepth with SiLog | | ✓ | | | | | | |
| + EST | | ✓ | ✓ | | | | | |
| + Metric loss | | ✓ | ✓ | ✓ | | | | |
| + Edge loss | | ✓ | ✓ | ✓ | ✓ | | | |
| + Multi-level temp. | | ✓ | ✓ | ✓ | ✓ | ✓ | | |
| + Multi-scale sup. | | ✓ | ✓ | ✓ | ✓ | ✓ | ✓ | |
| + Temporal reg. (EndoStreamDepth) | | ✓ | ✓ | ✓ | ✓ | ✓ | ✓ | ✓ |

Tab. 10 shows the ablation settings in our experiments, which are shown in Tab. 1.

