# OpenReview forum: "EndoStreamDepth: Temporally Consistent Monocular Depth Estimation for Endoscopic Video Streams"
_MIDL.io/2026/Conference — MIDL 2026 Poster_

### Official Review · Reviewer_WFDk · 2026-01-06

**Confidence:** 5
**Preliminary Rating:** 4
**Final Rating:** 5

**Summary:**

This paper introduces a streaming monocular depth estimation framework for endoscopic video, aiming to produce accurate metric depth, sharp anatomical boundaries, and temporally consistent predictions at real-time throughput.  Experiments on C3VD (including an in-distribution split and a domain-shift split) and SimCol3D show large gains over strong baselines (DepthAnything v2, EndoOmni, FlashDepth), improved boundary F1, and reduced temporal variance, with a reported runtime around 24 FPS on C3VD split 1.

**Strengths:**

1.	The three-part design (EST + single-frame foundation backbone + hierarchical multi-level temporal modules) is logically motivated, and each component addresses a concrete endoscopy-specific failure mode (lighting artifacts, rapid rotations, flicker).

2.	Using SiLog plus a log-metric loss and an edge gradient loss directly targets near-range accuracy and boundary sharpness—critical for endoscopic scenes that often contain low texture and subtle depth discontinuities.

**Weaknesses:**

1.	Some ablation results are counterintuitive (e.g., multi-level temporal modules reduce errors but slightly decrease boundary F1 and may introduce long-range flicker, as mentioned in the text). This warrants deeper analysis of the accuracy–sharpness–stability trade-off and when the modules help or hurt.

2.	The paper lacks ablation studies on: (i) the number of temporal levels, (ii) the number of Mamba blocks per level, and (iii) the temporal window size used during training (the paper mentions a window size of 5).

3.	Please report more detailed latency and memory usage, e.g., peak GPU memory and per-frame inference latency.

4.	Statistical significance analysis is missing. Please report mean ± standard deviation (and ideally statistical tests where appropriate).

**Detailed Comments:**

See weakness.

**Justification Of Final Rating:**

The authors have addressed all my concerns. Even though another reviewer still recommends borderline, I would like to strongly recommend this paper. Therefore, I would like to change my final rating from 4 to 5.

**Justification Of The Preliminary Rating:**

This paper introduces a streaming monocular depth estimation framework for endoscopic video, aiming to produce accurate metric depth, sharp anatomical boundaries, and temporally consistent predictions at real-time throughput.  Experiments on C3VD (including an in-distribution split and a domain-shift split) and SimCol3D show large gains over strong baselines (DepthAnything v2, EndoOmni, FlashDepth), improved boundary F1, and reduced temporal variance, with a reported runtime around 24 FPS on C3VD split 1. However, there are still some concerns listed in weakness. Therefore, I suggest weak accept.

**Questions To Address In The Rebuttal:**

See weakness.

---

> ### Author Response · Authors · 2026-01-25
>
> We thank reviewer’s comments.
>
> **Counterintuitive ablation results (fixed in main text).** We agree that the sentence was misleading due to misplacement. We have corrected it in the revised version.
>
> Regarding the slight decrease in boundary F1 after adding multi-level temporal modules, this is an expected accuracy vs. temporal trade-off. Temporal propagation operates at a coarse scale on high-level features, which carry good global context but limited fine details. Without an explicit constraint to preserve such fine details, the decoded depth maps can become over-smoothed, slightly reducing boundary F1 (0.135 to 0.123). This effect is also reflected in the near-range metrics, which slightly degrade after adding temporal modules (RMSE-near: 1.628 to 1.652). At the same time, global accuracy improves, as reflected by lower relative-scale and distance errors (SqRel: 0.387 to 0.379; RMSE: 2.917 to 2.810; MAE: 1.774 to 1.748), with larger improvements in mid/far ranges (RMSE-mid: 3.746 to 3.513; RMSE-far: 10.124 to 9.679). We define near, mid, and far depth ranges as <3, 3–7, and >7 mm.
>
> **Mamba ablation (added in main text).** Placing Mamba only at mid-levels [2,3] loses fine-grained boundary information. Using only the finest level [1] preserves local details but lacks global temporal context, leading to poor temporal stability. Missing the coarsest level limits accuracy for distance metrics due to missing global information. Our 4-level design captures complementary information across scales: coarse levels for global context and fine levels for boundary sharpness, achieving the best RMSE, F, and temporal stability.
>
> | | L1 | L2 | L3 | L4 | $\delta_1$ | AbsRel$ $ | RMSE | L1 | F1 | F-Var. |
> |---------|:--:|:--:|:--:|:--:|:--:|:--:|:--:|:--:|:--:|:--:|
> | Mid-level only | | ✓ | ✓ | | 0.952 | 0.086 | 2.996 | 1.869 | 0.117 | 0.0129 |
> | Fine only | ✓ | | | | 0.953 | 0.105 | 2.917 | **1.774** | 0.135 | 0.0137 |
> | Fine+mid | ✓ | ✓ |  | |**0.954**|0.087|2.975|1.817 |0.124 |0.0121 |
> | Fine+mid | ✓ | ✓ | ✓ | | 0.946 | 0.087 | 2.902 | 1.846 | 0.134 | 0.0124 |
> | full | ✓ | ✓ | ✓ | ✓ | 0.952 | **0.085** | **2.739** | 1.780 | **0.143** | **0.0123**|
> L1 denotes the finest level.
>
>
>
> **Window size analysis (added in main text).** Shorter temporal windows (Window=3) exhibit higher frame variance due to limited temporal context for enforcing consistency. Longer windows (Window=10) improve frame variance but blur depth boundaries through excessive temporal aggregation, resulting in lower F1. Our method with window length 5 achieves the optimal balance, maintaining low frame variance while preserving sharp boundaries.
>
> | | $\delta_1 $ | AbsRel| RMSE | L1 | F1 | Frame Var. |
> |---|:--:|:--:|:--:|:--:|:--:|:--:|
> | Window=3 | 0.948 | 0.090 | 2.945 | 1.907 | 0.131 | 0.0142 |
> | Window=5 (recommended) | **0.952** | 0.085 | **2.739** | **1.780** | **0.143** | **0.0123** |
> | Window=10 | 0.948 | **0.084** | 2.785 | 1.790 | 0.106 | 0.0125 |
>
> **Runtime (added Appendix G).**
>
> All benchmarks were conducted on a single NVIDIA RTX A6000 GPU at 518×518 resolution. We used bfloat16 mixed precision and torch.compile optimization. We skip the first 5 frames as warmup and measure steady-state performance on the remaining frames. We additionally benchmark up to 3000 frames, exceeding the maximum sequence length in our dataset (Tab. 6) to assess scalability.
> - **Constant inference speed:** Our model achieves a consistent per-frame latency of approximately 40 ms regardless of video length, demonstrating the low computational overhead of our Mamba-based temporal module.
> - **Linear memory scaling:** Peak GPU memory consists of a fixed base cost (2.5 GB for model weights and buffers) plus an incremental cost of approximately 4 MB per frame for temporal state. This predictable memory footprint enables practitioners to estimate memory requirements based on their target video length.
>
> | frames | Latency (ms) | FPS | Peak GPU Memory (GB) |
> |:--:|:--:|:--:|:--:|
> | 100 | 40.2 | 24.9 | 2.90 |
> | 500 | 40.0 | 25.0 | 4.50 |
> | 1000 | 40.3 | 24.8 | 6.50 |
> | 3000 | 40.7 | 24.6 | 14.50|
>
>
> **Statistical significance analysis (added Appendix D).** Our method achieves statistically significant improvements over both baselines on 5 of 7 metrics, indicating more robust performance across different test sequences.
>
> | Metrics | Metric DAv2 + EST | FlashDepth + SiLog + EST | Ours |
> |---------|:--:|:--:|:--:|
> | $\delta_1$ | 0.948±0.052 | 0.952±0.046 | 0.952±0.047 |
> | AbsRel | 0.109±0.025 | 0.109±0.021 | **0.085±0.021***|
> | SqRel | 0.402±0.135 | 0.395±0.100 | **0.246±0.109***|
> | RMSE | 3.081±0.528 | 3.023±0.552 | **2.739±0.442***|
> | RMSE log | 0.122±0.025 | 0.122±0.021 | **0.107±0.023***|
> | L1| 1.928±0.362 | 1.872±0.303 | **1.780±0.300***|
> | F1 | 0.114±0.045 | 0.134±0.069 | 0.143±0.059 |
>
> *p<.05, **p<.01, ***p<.001 (paired t-test for both methods).

---

### Official Review · Reviewer_kakV · 2026-01-09

**Confidence:** 4
**Preliminary Rating:** 3

**Summary:**

This paper presents a method for on-the-fly depth estimation from monocular endoscopic vision. The paper advocates interframe improvements using a Mamba module and makes use of scale invariance. The method is trained and tested on 2 publicly available synthetic datasets, and it is compared to some of the many existing methods. The results are encouraging.

**Strengths:**

The paper reads well, it concerns an important step in endoscopic procedures, and the results are encouraging

...................................................................................................

**Weaknesses:**

I'm not an expert on depth reconstruction in endoscopic streams, but a quick search shows that the field is strong and large, and the paper does not instill trust in being state-of-the-art. Further, since many methods seem to already be performing well, I'm missing a discussion on the required precision wrt. an end-goal, i.e., is the state-of-the-art a stopping block for navigation of cancer detection, etc. Wrt. the mathematical content, (1) is presented, but I did not understand why the -0.5 coefficient was optimal for the problem to be solved. Btw., this looks intriguingly similar to a variance estimation equation of log (D/\hat{D}), is this a coincidence? Eqn (2) is pitched as scale-variant in contrast to (1), which is the opposite. I suggest the authors consider the combined terms (log D/\hat{D}), which will show that (2) is indeed also scale-invariant. Finally, (3) includes the derivative of log D, which implies that the absolute gradient is scaled by depth. I'm missing a discussion on the influence of this on the method. Finally, a coefficient in (6) is set to 0.01 without a discussion.

**Detailed Comments:**

See above

**Justification Of The Preliminary Rating:**

State-of-the-art is uncertain, justification is weak, mathematical treatment is somewhat lacking quality.

..................................................................................................

**Questions To Address In The Rebuttal:**

See above

---

> ### Author Response · Authors · 2026-01-25
>
> We thank the reviewer for the detailed comments and perspective
>
> **Problem setting and performance.** We recognize the reviewer's concern about state-of-the-art comparisons. Depth estimation has several distinct problem settings:
> - **Single-frame methods:** Process individual images independently. They achieve good per-frame accuracy but produce temporally inconsistent ("flickering") predictions when applied to video
> - **Batched video methods:** Require multiple frames as simultaneous input (e.g., 32 frames). This introduces latency and prevents true real-time streaming, making them unsuitable for surgical robotics where frame-by-frame feedback is essential
> - **Stereo methods:** Require binocular cameras. Most clinical endoscopes are monocular, making these methods inapplicable
> - **Diffusion-based methods:** Produce high-quality depth but require iterative denoising (typically 10–50 steps), resulting in longer inference time not suitable for real-time applications
>
> Our work processes frames sequentially (one at a time) while enforcing temporal consistency through recurrent Mamba modules. This enables: (1) real-time throughput (24 FPS), (2) arbitrarily long video sequences (no batching), and (3) temporally stable predictions. Among prior works, FlashDepth is the most comparable method targeting causal real-time streaming; we outperform it by a large margin (Tab. 1, Fig. 4)
>
> **State-of-the-art performance.** Beyond C3VD split one, where we compared recent state-of-the-art methods, we validate on two public benchmarks:
> - **C3VD Split 2:** Our method achieves the best SqRel (0.11) and RMSE (1.72mm), outperforming PPSNet-Student (SqRel: 0.14, RMSE: 2.06mm), the previous state-of-the-art that uses a teacher-student framework with additional self-supervised learning
> - **SimCol3D challenge:** Our approach outperforms all four winning entries on L1 (0.087 vs. 0.099), RMSE (0.126 vs. 0.141), and AbsRel (0.023 vs. 0.025)
>
>
> **Required precision.** The reviewer is correct that existing single-frame methods (e.g., DepthAnything) can achieve good per-frame accuracy. However, they have two limitations when applied to endoscopy: (1) they flicker across frames, and (2) their performance is suboptimal for endoscopic scenes due to domain gap. Even EndoOmni, which is specifically designed for endoscopy, achieves similar performance to DepthAnything (Tab. 1), indicating that simply finetuning foundation models to endoscopy is insufficient
>
> EndoStreamDepth addresses both limitations. We outperform all single-frame and video-based state-of-the-art methods on per-frame metrics (Tab. 1), while also achieving superior temporal stability (Fig. 4)
>
> For closed-loop robotic control, both are essential. When depth models are applied to video, flickering causes depth values to jump unpredictably between frames. A surgical robot using flickering depth would receive unstable feedback, potentially causing erratic motion or unsafe tool-tissue interaction. The control loop requires depth that is not only accurate but also stable across frames
>
> Our multi-level temporal Mamba and temporal regularization specifically target this issue. Our concurrent work PERSEUS [1] shows that integrating temporally stable MDE into SLAM reduces reconstruction noise and drift. For our autonomous resection workflows [2,3] that require real-time intra-operative feedback, consistent streaming depth (24 FPS) enables reliable closed-loop control where flickering single-frame methods would fail
>
> 1. PERSEUS: Perception with Semantic Endoscopic Understanding and SLAM.
> 2. From monocular vision to autonomous action: Guiding tumor resection via 3D reconstruction.
> 3. A Supervised Autonomous Resection and Retraction Framework for Transurethral Enucleation of the Prostatic Median Lobe
>
>
> **Detailed comments**
>
> 1. **Eq. (1): the −0.5 term and relation to variance.** Let $d_i = \log D_i - \log \hat{D}_i$. Our Eq. (1) follows the standard SiLog loss (Depth Anything v2). The −0.5 term reduces sensitivity to global scale offset, and 0.5 is a conventional setting rather than a tuned hyperparameter. The resemblance to variance is intentional: $\text{Var}(d) = \mathbb{E}[d^2] - \mathbb{E}[d]^2$, and SiLog uses this structure with a controllable weight
>
> 2. **Eq. (2): scale sensitivity.** In supervised training, the ground truth $D$ is fixed. If the prediction $\hat{D}$ has a systematic scale bias, Eq. (2) will penalize it
>
> 3. **Eq. (3): effect of $\nabla \log D$.** $\nabla \log D = \nabla D / D$, which corresponds to a relative gradient. This emphasizes boundary alignment while reducing domination by large absolute depth values. It encourages sharper anatomical boundaries and is more robust across depth ranges
>
> 4. **Eq. (6): temporal coefficient 0.01** $\mathcal{L}_{\text{temp}}$ is applied to per-video normalized depth, so its magnitude is not directly comparable to supervised metric losses. We set weight 0.01 to reduce frame-to-frame fluctuations without over-smoothing spatial details

---

### Official Review · Reviewer_QMzP · 2026-01-10

**Confidence:** 4
**Preliminary Rating:** 4
**Final Rating:** 4

**Summary:**

1. The paper presents **EndoStreamDepth**, a monocular depth estimation framework designed for **real-time endoscopic** video streams that emphasizes spatial accuracy, sharp anatomical boundaries, and temporal consistency.

2. The method combines a strong single-frame depth backbone with hierarchical, multi-level temporal modeling using Mamba modules, enabling streaming inference without requiring batched frames

3. The authors also present endoscopy-specific adjustments to enhance resilience to geometric and photometric fluctuations, alongside extensive supervision incorporating scale-invariant, metric, edge-aware, multi-scale, and self-supervised temporal losses.

4. Extensive experiments on public phantom and simulated datasets demonstrate improved depth accuracy, boundary sharpness, and reduced temporal flicker while maintaining real-time throughput, supporting the method’s suitability for downstream robotic and image-guided applications.

**Strengths:**

Strengths

1. **Clear Problem Statement:** The paper targets a well-defined and important challenge which is directly relevant to medical robotics and image-guided interventions.
2. **Streaming-friendly design:** Unlike many prior video depth methods that rely on batched multi-frame inputs, EndoStreamDepth processes frames sequentially, making it suitable for low-latency deployment.
3. **Strong temporal modeling:** The hierarchical multi-level Mamba-based temporal architecture is well motivated and effectively reduces frame-to-frame flickering, as supported by both quantitative variance metrics and qualitative visualizations.
4. **EST Model:** The proposed Endoscopy-Specific Transformation (EST) thoughtfully models common geometric and photometric artifacts in endoscopic imaging and is shown to substantially improve robustness.
5. **Thorough experimental evaluation:** The paper reports results across multiple datasets, metrics, ablations, and runtime analyses, and provides public code, supporting reproducibility.

**Weaknesses:**

**Evaluation limited to phantom and simulated datasets**: While appropriate for controlled benchmarking, the absence of in-vivo clinical data limits conclusions about robustness under real procedural conditions (e.g., fluids, extreme non-rigid motion).

**Incremental architectural novelty**: Several components (ViT+DPT backbone, Mamba temporal modules, standard depth losses) are adaptations of existing ideas rather than fundamentally new, although their integration is well executed.

**Runtime trade-offs**: The method is slower than the closest streaming baseline, and while still real-time, a more detailed discussion of deployment constraints (e.g., latency budgets in robotic systems) would strengthen the practical narrative.

*Finally, the paper can go further and empirically demonstrate improvements in navigation, control, or task performance.*

**Detailed Comments:**

1. The paper is generally well written; minor redundancy between Sections 2.2 and 2.3 could be streamlined.
2. In the paper - choice of Mamba over alternative temporal models is reasonable. Can a a short discussion on why other lightweight recurrent mechanisms were not considered be added?
3. The EST pipeline is effective; a brief ablation isolating geometric vs photometric transformations could further clarify their individual contributions.

**Justification Of Final Rating:**

I thank the authors for the detailed and constructive rebuttal. The responses address my main questions clearly.

Overall, the rebuttal satisfactorily resolves my major concerns. The remaining limitations are clearly articulated and do not detract from the paper’s core contribution as a privacy-preserving and clinically motivated synthetic data framework. Additionally, thanks for the quick turnaround in updating the manuscripts.

**Justification Of The Preliminary Rating:**

1. This paper presents a solid and carefully engineered solution to real-time monocular depth estimation for endoscopic video.
2. The streaming architecture, hierarchical temporal modeling, and comprehensive supervision are well motivated and convincingly validated on public benchmarks.
3. While the work is largely incremental in terms of individual components and lacks in-vivo evaluation, the overall system integration and empirical performance represent a meaningful contribution to the field.


The paper meets the standards for acceptance at MIDL and will be of interest to researchers working on medical video understanding and robotic perception.

**Questions To Address In The Rebuttal:**

1. How does *EndoStreamDepth* perform under severe non-rigid motion or heavy fluid artifacts, and are there plans for preliminary in-vivo validation?
2. How sensitive is the temporal regularization to sudden camera reversals or occlusions, which are common in clinical procedures?
3. Could the proposed EST be transferred directly to other endoscopic modalities (or) is re-tuning required?

---

> ### Author Response · Authors · 2026-01-25
>
> We thank the reviewer for the thoughtful comments.
>
> **In-vivo validation.** We acknowledge that evaluation on in-vivo data is an important next step. Our current phantom and simulation benchmarks were chosen because they provide ground-truth depth, which is essential for quantitative evaluation. However, obtaining such ground truth in live procedures remains an open challenge for the community, since we cannot, e.g., place trackers in the body.
>
> **Novelty.** We respectfully note that while individual components build on prior work, as detailed in the manuscript (p2), our contributions are:
> - **EST:** A simple but effective transformation strategy specifically designed for endoscopy, which has not been systematically studied in prior depth estimation work. Table 1 shows its effectiveness.
> - **Comprehensive supervision:** The combination of SiLog, metric, and edge losses is carefully designed for endoscopy's unique challenges (near-range accuracy, boundary sharpness, temporal stability).
> - **Multi-level temporal Mamba:** FlashDepth uses a single temporal module; we extend this to a hierarchical 4-level design with multi-scale supervision and temporal regularization. Despite our superior depth accuracy, this component primarily targets temporal stability rather than directly optimizing per-frame accuracy, so improved temporal scores can occur even under imperfect frame-wise predictions. We observe reduced variance on 8/9 videos (Fig. 4).
>
> Our method achieves state-of-the-art results on two public benchmarks. More importantly, to the best of our knowledge, this is the first streaming depth estimation work for endoscopy, and we believe it can serve as a strong baseline for future work and benchmarking, especially for real-time depth estimation.
>
> **Runtime (added Appendix G).** Our method achieves 24 FPS, which exceeds the 20 FPS threshold typically required for real-time surgical feedback systems [1]. While FlashDepth is faster, our method provides substantially better accuracy and temporal stability. For robotic applications where depth accuracy directly impacts control quality, we believe this trade-off is favorable.
>
> [1] Deep learning in real-time image-guided surgery: a systematic review
>
> **Camera reversals (added Appendix E).** Fig. 6 (Video 2b) shows a forward-to-backward camera motion sequence. Despite the sudden direction change, our method maintains stable depth predictions without noticeable flickering. The multi-level temporal Mamba uses recurrent hidden states that adapt to motion changes, rather than assuming smooth camera trajectories. The frame variance remains low (0.0127), indicating robustness to such reversals.
>
> **Occlusions (Appendix A).** Our EST augmentation includes random fog and blur (Appendix A), simulating partial occlusions. Training with these augmentations improves robustness to temporary occlusions in real procedures.
>
> **Non-rigid motion and fluid artifacts.** Our datasets do not contain severe non-rigid motion or heavy fluid artifacts. EST transformation provides partial robustness to visual disturbances, but real clinical scenarios with fluid or tissue deformation remain challenging. This is an important direction for future work.
>
> **Temporal module choice (added Appendix E).** We choose a Mamba-based streaming temporal module to satisfy the real-time, per-frame inference constraint with a persistent state. This follows FlashDepth, which shows vanilla Mamba is sufficient. Building on this validated design, our contribution focuses on EST, multi-level temporal, and an additional scale/shift-invariant temporal regularization to reduce flickering.
>
> **EST generalization (added in main text).** EST is a modality-agnostic, stochastic augmentation (no learned parameters), so it can be applied directly to other endoscopic modalities.
>
> **EST ablation (added in main text).**
>
> | | $\delta_1$ | $\delta_{1,near}$ | RMSE | RMSE$_{near}$ | L1 | L1$_{near}$ | AbsRel |
> |---------|------------|-------------------|------|---------------|-----|-------------|--------|
> | FlashDepth + SiLog | 0.853 | 0.831 | 3.774 | 2.789 | 2.695 | 2.197 | 0.139 |
> | + photometry | 0.835 | 0.821 | 4.868 | 3.180 | 3.160 | 2.336 | 0.156 |
> | + geometry | 0.922 | 0.911 | 3.083 | 2.010 | 1.962 | 1.521 | 0.099 |
> | + EST | 0.952 | 0.946 | 3.023 | 1.715 | 1.872 | 1.284 | 0.108 |
>
> "near" denotes depth < 3mm.
>
> Photometric transformations alone reduce performance, while geometric transformations improve performance across metrics. Importantly, full EST yields the largest gains in the near range, which is critical for surgical robotics.
>
> **Improvements in navigation.** Our concurrent work PERSEUS [1] empirically demonstrates that depth estimation quality directly impacts downstream task performance. Integrating MDE priors into DROID-SLAM reduces reconstruction noise and drift, producing more accurate 3D maps.
> [1] PERSEUS: Perception with Semantic Endoscopic Understanding and SLAM

---

> > ### Comment · Reviewer_QMzP · 2026-01-25
> > **Follow-up After Rebuttal**
> >
> > 1. The revised manuscript and supplementary material convincingly address the main concerns raised in my review.
> >     * In particular, the added ablation studies clearly isolate the contributions of EST, multi-level temporal Mamba modeling, and temporal regularization, strengthening the paper’s technical positioning.
> >     * The EST component is now well-motivated and empirically justified, especially for near-range accuracy critical to robotic applications.
> >
> >
> > 2. I also appreciate the added discussion and evidence regarding robustness to camera reversals and occlusions, as well as the new runtime and memory analysis, which makes the real-time deployment claims much more concrete.
> >
> > The authors appropriately acknowledge the limitations of phantom and simulated data and clearly frame in-vivo validation and more challenging clinical artifacts as future work, which I find reasonable given current ground-truth constraints.
> >
> > Overall, the rebuttal significantly improves clarity, rigor, and practical relevance.
> >
> > I encourage the authors to make minor final edits to explicitly emphasize the controlled evaluation setting early in the paper and to briefly contextualize the assumed latency requirements. With these minor refinements, I believe the paper meets the standards for acceptance and will be a valuable contribution to the community.

---

> > ### Author Response · Authors · 2026-01-26
> >
> > Thank you for the quick follow-up and for the supportive comments. We appreciate your suggestions. As the Program Chairs note that the rebuttal PDF is final during the discussion period, we cannot update the manuscript at this stage, but we will incorporate these two points in the camera-ready version. Specifically, we will: (1) explicitly emphasize early in the paper that our quantitative evaluation is conducted on phantom and simulated data with ground-truth depth, and (2) briefly contextualize the suggested latency requirements [1]. Thank you again.
> >
> > [1] Deep learning in real-time image-guided surgery: a systematic review

---

> > ### Author Response · Authors · 2026-01-27
> >
> > As the program chair has extended the revision draft deadline, we have revised our submission and uploaded the updated manuscript. We explicitly emphasize the controlled evaluation in the Abstract and Introduction. We also briefly contextualize the assumed latency requirement (20 FPS) in the Introduction, Results, and Appendix G. Thanks again for your comments.

---

> ### Author Response · Authors · 2026-01-28
> **Clarification on two statements in the final justification**
>
> Thank you for the final assessment and for taking the time to read our rebuttal. However, we do not fully understand parts of your **Justification of Final Rating**, and we are concerned that some statements may not correspond to our submission.
>
> We would like to flag two points:
>
> **(1) “Flow-based refinement.”** Our method does not use optical flow, warping, or any flow-based refinement stage. The temporal improvements in EndoStreamDepth come from the streaming Mamba-based temporal module (hidden-state propagation across frames), together with EST and our training objectives, as supported by the ablations and supplementary material.
>
> **(2) “Horizontal jerk / pendular nystagmus” and “waveform distribution analysis / Wasserstein distance.”** Our paper does not study nystagmus types, does not analyze waveform distributions, and does not introduce Wasserstein distance (or related divergence measures) as an evaluation metric. Our scope is endoscopic monocular depth estimation and temporal stability/consistency of depth sequences.
>
> Could you please clarify how these two statements relate to our submission (e.g., which components or experiments you are referring to)? This would help us ensure we are interpreting your final justification correctly. Thank you!

---

> > ### Comment · Reviewer_QMzP · 2026-01-28
> > **Quick clarification**
> >
> > Thank you for flagging this — you are absolutely right, and I appreciate the opportunity to clarify.
> >
> > Flow-based refinement. My wording here was imprecise. EndoStreamDepth does not use optical flow, warping, or any explicit flow-based refinement. What I intended to refer to was the streaming temporal refinement achieved via the recurrent Mamba hidden-state propagation together with temporal regularization, which improves temporal stability across frames. I apologize for the inaccurate phrasing.
> >
> > Nystagmus, waveform analysis, and Wasserstein distance. You are correct that these concepts are not part of your paper. Their mention in my justification was an error and does not correspond to any component, experiment, or claim in your submission. Please disregard these statements entirely.
> >
> > To be clear, these wording issues do not affect my final assessment or rating. My evaluation is based on the streaming Mamba-based temporal modeling, EST, hierarchical supervision, and the empirical temporal stability results you present.
> >
> > I apologize for the confusion caused by these misstatements, and thank you again for your careful reading and constructive engagement during the discussion.

---

### Author Rebuttal · Authors · 2026-01-25

**Rebuttal:**

We thank the reviewers for their detailed and constructive comments. We have addressed them point-by-point and uploaded the revised manuscript.

**Supporting Material:**

/attachment/fcec58c7c9c598efab3af9b4a12af0bd5572f7de.pdf

---

### Meta-Review · Area_Chair_Aapb · 2026-02-07

**Recommendation:** Accept (Poster)
**Confidence:** 4

**Metareview:**

The manuscript received mixed reviews initially, and underwent quite substantial revisions that convincingly addressed two of the three reviewers' concerns. One reviewer did not update their score, though from my read of the rebuttal and the thorough discussion that this manuscript enjoyed, I am in agreement with the positive assessment.

---

### Decision · Program_Chairs · 2026-02-13

Accept (Poster)